# Bilateral Fuchs’ Superficial Marginal Keratitis Diagnosis and Treatment

**DOI:** 10.3390/life14121644

**Published:** 2024-12-11

**Authors:** Shiori Harada, Yasser Helmy Mohamed, Mao Kusano, Daisuke Inoue, Masafumi Uematsu

**Affiliations:** Department of Ophthalmology and Visual Sciences, Graduate School of Biomedical Sciences, Nagasaki University, Nagasaki 852-8501, Japan; haradashiori1211@yahoo.co.jp (S.H.); maok@nagasaki-u.ac.jp (M.K.); d.i.private.3@gmail.com (D.I.); uematsu1124@outlook.jp (M.U.)

**Keywords:** Fuchs’ superficial marginal keratitis, pseudopteryguim, irregular astigmatism, corneal thinning

## Abstract

In this study, we reported two patients with clinical pictures compatible with the diagnosis of bilateral Fuchs’ superficial marginal keratitis (FSMK) treated with surgical treatment and anti-inflammatory drugs. The cases suffered from bilateral photophopia, blurred vision, and pseudopterygium with normal intraocular pressure (IOP). Pseudopterygia extended from temporal and nasal sides and had a gray line between the corneal normal epithelium with no lipid deposits. The first case had a bilateral high mixed astigmatic error and the second had dry eye symptoms. No abnormalities, including systemic inflammatory disease, were found in the laboratory investigations. The first case had bilateral pseudopterygium excision, mitomycin C (MMC) application, and pedicled conjunctival flap transplantation. The patient was followed regularly, and her condition was stable without any recurrence or deterioration in the ocular findings. The second case had conjunctival resection + amniotic membrane transplantation + MMC application six times on the right eye and three times on the left eye during 4 years and suffered recurrences after each operation. Her visual acuity severely decreased with the elevated IOP of both eyes. The patient refused to do any further surgical intervention. Despite its rarity, FSMK should be considered when peripheral corneal infiltration, corneal thinning, and pseudopterygia are observed in both eyes.

## 1. Introduction

Fuchs’ superficial marginal keratitis (FSMK) is a rare disorder that was first documented by Von Arlt in l88l and then described in further detail by Fuchs in 1895 [1].

FSMK is a rare condition marked by episodic ocular inflammation, marginal corneal infiltrates, and the progressive thinning of the corneal stroma. A pseudopterygium may develop over the corneal thinning. ln advanced cases, extreme stromal loss may lead to traumatic or even spontaneous perforation [2]. This disease of unknown etiology usually starts as a superficial marginal keratitis that spreads over the cornea in an irregular, nonuniform manner [3,4].

Because the disease is rare, only a few reports in the literature have described it. We report two patients with clinical pictures compatible with a diagnosis of bilateral Fuchs’ superficial marginal keratitis treated with surgical treatment and anti-inflammatory drugs. This study was approved by the Institutional Review Board of Nagasaki University Hospital (approval number 24021928) and the patients involved in the study provided informed consent.

## 2. Case Report

### 2.1. Case 1

A 47-year-old Japanese woman with no underlying disease was referred to Nagasaki University Hospital in April 2022 by ophthalmology clinic suffering binocular blurred vision and photophobia for the past 6 years. Her best corrected visual acuity (BCVA) at the first visit was 0.2 (decimal correction) in right eye (refraction was +11.0 D sphere, −5.0 D axis 170 cylinder) and 0.4 in the left eye (refraction was +10.0 D sphere, −5.0 D axis 170 cylinder) with normal intraocular pressure (IOP) in both eyes. Both eyes had circumferential corneal infiltrations and pseudopterygia extending from the temporal and nasal sides and had a gray line between the corneal normal epithelium with no lipid deposits (Figure 1A,E). Anterior segment optical coherence tomography (AS-OCT) revealed bilateral pseudopterygium and mild corneal thinning in the same area (Figure 1B,F). Corneal elevation topography revealed binocular irregular astigmatism (Figure 1C,G). No abnormalities, including systemic inflammatory disease, were found in the laboratory investigations. A diagnosis of FSMK was made.

In June 2022, bilateral pseudopterygium excision, mitomycin C (MMC) application, and pedicled conjunctival flap transplantation were performed (Figure 1D,H). Postoperative treatment includes topical antibiotics and topical steroids for one month (four times/day). Topical antibiotics were discontinued one month after surgery, but weak topical corticosteroids continued to control congestion of the conjunctiva. The oral administration of cyclosporine was started one month after surgery to suppress postoperative inflammation and prevent recurrence to date (Dose = 175 mg/day which control the trough value between 70 and 100 ng/mL).

The patient was followed every 2–3 months, and her condition was stable without any recurrence or deterioration in the ocular findings until her last visit in December 2023. The last follow up picture of her eyes shown in (Figure 2) and her last BCVA was 1.2 in right eye (refraction was +4.0 D sphere, −2.0 D axis 150) and 1.2 in left eye.

### 2.2. Case 2

A 28-year-old female suffered from dryness and photophobia in her both eyes for a few years and was diagnosed with dry eye and pseudopterygium by local ophthalmologist. She was treated with eye drops containing lubricants and topical steroids. When she was 29 years old (November 2011), she was referred to Nagasaki University Hospital for better evaluation and treatment. Her BCVA was 0.6 in right eye (refraction −1.25 D sphere) and 0.9 in left eye with normal IOP in both eyes. There was hyperemia in the bulbar conjunctiva of both palpebral fissures, and the conjunctival epithelium had invaded the cornea circumferentially (Figure 3A,E). She had no conjunctival adhesions. Punctate superficial keratopathy was observed in the lower halves of both corneas, and conjunctival invasion was observed on her nasal and temporal corneas (Figure 3B,F). There were no major abnormalities in corneal elevation topography (Figure 3C,G).

The anterior chambers had normal depth in both eyes with no signs of inflammation, clear lens, and normal fundus of both eyes. Her underlying illnesses included obsessive-compulsive disorder [patient under selective serotonin reuptake inhibitors (SSRIs)], Sjögren’s syndrome, and allergic rhinitis. She did not have rheumatoid arthritis.

She was treated with eye drops (lubricants, topical steroids, and tranilast) 4–6 times a day for both eyes with punctal plugs, but the conjunctival invasion gradually worsened. Between 2011 and 2015, the patient was followed every two months. She underwent the first conjunctival resection + MMC + amniotic membrane transplantation in her right and left eyes in June and July 2015, respectively (Figure 3D,H). She started oral prednisolone and cyclosporine postoperatively. Oral prednisolone 20 mg/day was started on the third day after right eye surgery and was tapered after 4 months. Oral prednisolone were used in the same regimen during the second and third perioperative periods. Also, oral cyclosporine 150 mg/day was started two weeks after the first right eye surgery. Cyclosporine is being controlled with a trough value of 70–100 ng/mL). The recurrence of the conjunctival invasion occurred after two months of surgery. The patient underwent conjunctival resection + amniotic membrane transplantation + MMC application (+keratoepithelioplasty on the fourth surgery on right eye) 6 times in the right eye and 3 times in the left eye between the ages of 33 and 37 years (2015–2019) and suffered recurrences after each operation. Both eyes showed progressive conjunctival invasion with repeated peripheral corneal ulcers (Figure 4).

In February 2019, her visual acuity had decreased to 0.3 in her right eye and 0.07 in her left eye. In addition, elevated IOP was observed both eyes (right = 21 mmHg and left = 26 mmHg), and glaucoma eye drops were prescribed. On her last visit in November 2023 (41 years), her visual acuity further deteriorated to 0.01 in both eyes and intraocular pressure in the right eye was 15 mmHg and 24 mmHg in the left eye with medication. At the last visit, she was using the following eye drops: tacrolimus 2 times a day, tranilast 4 times a day, lubricants 2 times a day, and glaucoma eye drops. Figure 5 shows the patient’s corneas on her last visit (November 2023). The patient refused to do any further surgical intervention, and she continued to have the same oral and eye drop treatments.

## 3. Discussion

FSMK is a rare entity characterized by episodic ocular inflammation with marginal corneal infiltrates and progressive marginal thinning of the corneal stroma [2]. It most commonly affects young adults between the second and 4th decades of life. It often has a chronic course with recurrent bouts of red eye, pain, and tearing, associated with a marginal stromal thinning, which is typically irregular in its depth and axial extension, delimited by a fine intraepithelial gray line on its advancing edge and without any accompanying lipid deposits [3,5]. The marginal thinning does not have a preferred limbal location and is frequently associated with a pseudopterygium [4,5]. Visual acuity can be compromised in advanced cases secondary to irregular astigmatism [6].

Given the striking similarity in clinical manifestations between our patients and the previously described FSMK cases, FSMK emerged as the most probable diagnosis.

The differential diagnosis of FSMK includes autoimmune peripheral ulcerative keratitis, Mooren’s ulcer, Terrien’s marginal degeneration (TMD), and limbal stem cell deficiency (LSCD). Autoimmune peripheral ulcerative keratitis was ruled out because various blood tests were negative for collagen disease and systemic inflammation. Mooren’s ulcer manifested with painful progressive corneal peripheral ulceration (epithelial defect stained with fluorescein) associated with intense limbal inflammation and swelling, were not identical with our cases.

TMD and FSMK are rare, slowly progressive corneal diseases that affect both eyes and have an unknown cause. They share several clinical signs. TMD is marked by the peripheral thinning of the cornea, typically starting in the upper half but sometimes beginning in the lower half [7]. For FSMK, an early biomicroscopic examination usually reveals peripheral corneal thinning in the lower and lower nasal segments. As a rule, in both diseases, the lesions on the two eyes are asymmetric, but peripheral corneal thinning is the main pathological sign [7]. FSMK is a rare disorder that shares several characteristics with TMD, to the extent that they can be considered a different manifestation of the same degenerative marginal corneal disease whose etiology is unknown [7].

FSMK and TMD are two conditions that primarily affect young and middle aged adults. They both involve paralimbal stromal thinning, typically occurring bilaterally but asymmetrically. While pseudo-pterygium can occur in either disease, it is more classically associated with FSMK. Additionally, both conditions can lead to spontaneous or traumatic corneal perforation [5,8]. Because of these similarities, some authors had suggested that both disorders are manifestations of the same disease [5,7]. FSMK differentiates from TMD in that it affects the limbus at any localization (no preference for superior cornea), presents with accompanying epithelial detects, has an epithelial delimitation gray line, and does not have lipid deposition [3,5].

LSCD can be distinguished from FSMK by its underlying cause, such as mechanical or chemical trauma or other known factors contributing to LSCD [8]. However, because FSMK gradually damages corneal limbal stem cells and causes conjunctivization as it progresses, the possibility that this rare disease is responsible for some LSCDs of unknown cause cannot be ruled out. Determining whether FSMK is a contributing factor to LSCDs will have to wait until more cases accumulate.

Progressive stromal thinning is a risk factor for traumatic and spontaneous perforation [3,9]. It may be difficult to determine the degree of thinning secondary to accompanying pseudopterygium. Care must be taken during surgical intervention because perforation may accidentally occur [9]. The recurrence of the disease has been noted to occur even after lamellar keratoplasty, although this may be peripheral to the border of the graft [2,10].

AS-OCT may be an invaluable preoperative tool as it will allow for the differentiation of the pseudopterygia tissue from the underlying thinned corneal tissue. Although topography may measure the cumulative thickness encompassing both pseudopterygia and corneal tissue, AS-OCT will allow differentiation based on cross-sectional appearance. Cheung et al., observed, that in their FSMK case, the nasal part of the cornea was the thinnest, measuring 34% of the central corneal thickness. However, they suggested that the superonasal quadrant might actually be the thinnest area and the site of the perforation. Interestingly, the AS-OCT images demonstrated thinning to be greatest in the mid-periphery [10]. AS-OCT made it possible to assess the thickness of the cornea more accurately in the affected area, which is important for choosing treatment tactics [7].

The marginal superficial limbal corneal degeneration and ulcers are characteristically demarcated from the central cornea by a curvilinear gray band [3]. The central cornea is generally clear until the late stages of the disease, when visual acuity deteriorates. This is a chronic disease with periods of remission and relapse accompanied by processive circumferential peripheral corneal thinning that may lead to perforation [9]. Pseudopterygia tend to grow over areas of corneal thinning [4].

Ellis studied two cases of bilateral FSMK and described the histopathology of the disease. One of the eyes showed epithelioid giant cells in the substantia propria of the cornea. Acute inflammatory cells were present most concentrated beneath the ulcerated areas [11]. The degeneration of conjunctival and corneal limbal areas were encountered similar to that of pterygia [11].

The pseudopterygium gradually extends onto the cornea over several years, avoiding the central cornea. Bierly et al. described significant thinning beneath the pseudopterygium in two out of three cases. This resulted in corneal perforation during pseudopterygium excision in one case and following blunt trauma in another. These complications highlight the necessity for special precautions when treating these patients [3].

FSMK treatment depends on its stage. Symptoms’ improvement with the use of lubricants [4], topical steroids [4,5], oral doxycycline [5], and vitamin C in the acute phase has been reported [5,6]. The active inflammatory process is well controlled by instillations of corticosteroids, but this group of drugs should be used judiciously, determining the duration and dose. If complaints of severe pain in the eye, severe corneal syndrome, and ineffectiveness of conservative therapy persisted, a perilimbal conjunctivotomy was performed, consisting in the separation of the conjunctiva along the entire circumference of the limbus of the affected eye [7].

Chronic topical corticosteroid treatment may have been effective in limiting the progression of the disease. While FSMK and TMD are usually not treated with long-term anti-inflammatory medications unless there are clinical signs of ocular inflammation, the early use of such therapy might help limit disease progression [5].

In cases of formed reverse astigmatism, it is possible to use scleral contact lenses, which can significantly improve visual acuity. In rare cases, from minimal or even no trauma in patients with TMD and FSMK, perforation is noted, the probability of which is about 15%, as well as the subtotal or total detachment of Descemet’s membrane, corneal edema, and the formation of corneoscleral or intracorneal cysts [7].

Cyanoacrylate has transient usefulness in corneal microperforations. There are a few case reports treated with penetrating keratoplasty [3], lamellar keratoplasty [2], amniotic membrane reconstruction [4], superficial keratectomy with conjunctival autograft [4,5], and corneoscleral lamellar patch graft [6].

Keenan et al. highlighted the difficulty in treating slowly progressive peripheral corneal thinning associated with FSMK. They observed progressive thinning despite treating with doxycycline and vitamin C [5]. This approach to treatment had theoretical benefits because in vitro studies have shown that doxycycline inhibits matrix metalloproteinase activity and ascorbate enhances collagen synthesis [12,13].

Pseudopterygia in FSMK may progress to threaten visual acuity. Surgical excision can be safe and can effectively improve vision on the condition that the patients are closely followed in the early postoperative period to notice the corneal infiltrates [7]. Recommended surgical treatment may entail combining superficial keratectomy with a conjunctival autograft or amniotic membrane transplantation to try to retard recurrent pseudopterygia formation through the same mechanism that prevents recurrence after the excision of traditional pterygia [4,14]. Kotecha and Raber noted that this treatment seemed to suppress the flare-ups of marginal keratitis. Given the role that the conjunctiva may play in inflammation, a conjunctival autograft in conjunction with a lamellar patch graft has also been recommended [4].

In severe cases, like our second case, it was difficult to treat even with multiple surgical treatments with long term use of oral cyclosporin. However, in the first case, surgery was performed early in the disease, prior to widespread conjunctival invasion. Oral cyclosporin was administered immediately post-surgery, and no deterioration in the corneal condition was observed for 1.5 years following the procedure. Our cases demonstrated the two extremes of the disease and their responses to treatment, making it difficult to predict treatment outcomes.

It is possible that the progression of FSMK may be suppressed by surgical treatment and strict inflammation control from the early stage of the disease.

## 4. Conclusions

We encountered two cases where refractory peripheral corneal infiltration and pseudopterygia were present in both eyes, leading to a strong suspicion of FSMK. Despite its rarity, FSMK should be considered when peripheral corneal infiltration, corneal thinning, and pseudopterygia are observed in both eyes. Although treating this condition is challenging, timely surgical and anti-inflammatory interventions may help manage the disease.

## Figures and Tables

**Figure 1 life-14-01644-f001:**
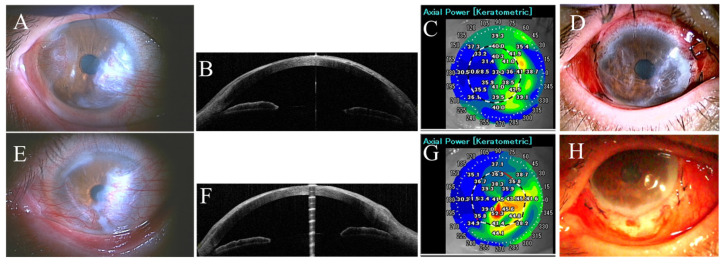
Slit lamp pictures of right (**A**) and left (**E**) eyes of the first case (first visit) showing circumferential corneal infiltrations and pseudopterygia extending from the temporal and nasal sides. Anterior segment optical coherence tomography revealed pseudopterygium and mild corneal thinning in both temporal and nasal sides of right (**B**) and left (**F**) eyes. Corneal elevation topography revealed binocular irregular astigmatism in the right and left eyes ((**C**,**G**) respectively). Slit lamp pictures of right (**D**) and left (**H**) eyes on the first postoperative day.

**Figure 2 life-14-01644-f002:**
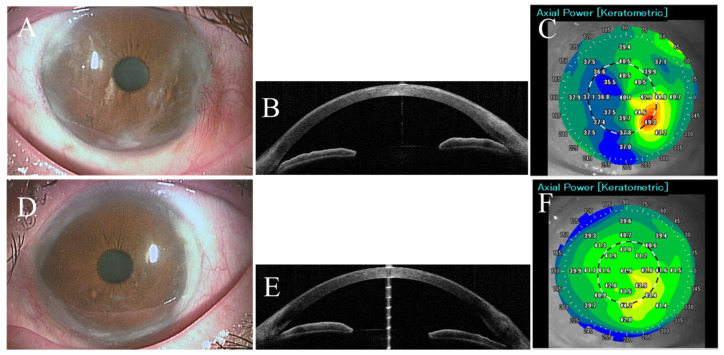
Slit lamp pictures of right (**A**) and left (**D**) eyes of the first case showing the postoperative removal of corneal pseudopterygia from the temporal and nasal sides. Anterior segment optical coherence tomography confirmed the postoperative removal of pseudopterygium of right (**B**) and left (**E**) eyes. Postoperative corneal elevation topography revealed binocular improvement in right and left eyes ((**C**,**F**) respectively).

**Figure 3 life-14-01644-f003:**
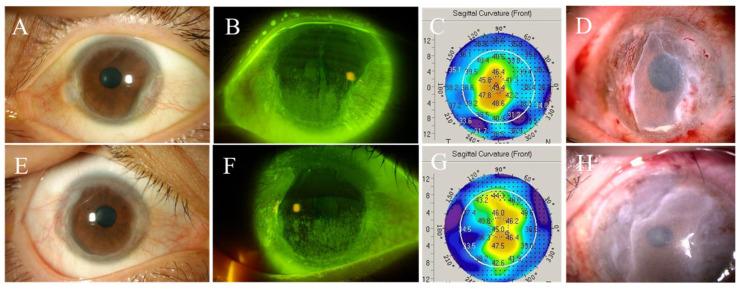
Slit lamp pictures of right (**A**) and left (**E**) eyes of the second case (first visit) showing circumferential corneal infiltrations and pseudopterygia extending from the temporal and nasal sides. Punctate superficial keratopathy was observed in the lower halves of right and left corneas ((**B**,**F**) respectively), and conjunctival invasion was observed on her nasal and temporal corneas. There were no major abnormalities in the corneal elevation topography of the right (**C**) and left (**G**) eyes. Slit lamp pictures of right (**D**) and left (**H**) eyes on the first postoperative day after first operation.

**Figure 4 life-14-01644-f004:**
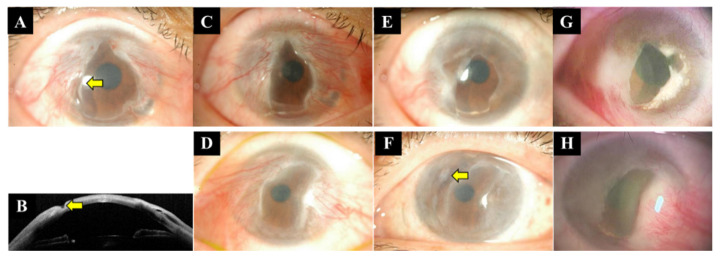
Slit lamp picture (**A**) and ocular coherent tomography (**B**) of the right eye of the second case at age of 32 years showing corneal ulceration and thinning (yellow arrow). Slit lamp pictures of the right (**C**) (after healing of ulceration) and left (**D**) eyes of the second case at the age of 32 years revealing progressive encroachment of psudopterygium on the corneal surface. Slit lamp pictures of right (**E**) and left (**F**) at the age of 34 years. Left corneal ulcer and thinning (yellow arrow) was shown. Slit lamp pictures of right (**G**) and left (**H**) at the age of 37 years.

**Figure 5 life-14-01644-f005:**
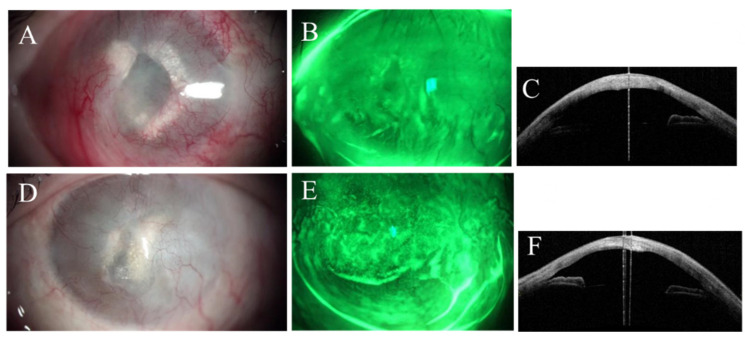
Slit lamp pictures of right (**A**) and left (**D**) eyes of the second patient in November 2023 (41 years) showing extensive encroachment of psudopterygium on the corneal surface. Slit lamp pictures of right (**B**) and left (**E**) eyes of the second patient upon her last visit showing extensive punctate superficial keratopathy. Ocular coherent tomography of right (**C**) and left (**F**) eyes of the second patient at her last visit showing extensive encroachment of psudopterygium on the corneal surface.

## Data Availability

The datasets used during the current study are available from the corresponding author upon reasonable request.

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
