# Peer review of "Bilateral Fuchs’ Superficial Marginal Keratitis Diagnosis and Treatment"

_life, 2024, doi:10.3390/life14121644_

Round 1

Reviewer 1 Report

Comments and Suggestions for Authors

1. It would be greatly appreciated if you could include a photograph taken immediately post-surgery or on the first day after surgery for the first patient. This would make it easier to understand the specific surgical techniques that were employed.

2. If possible, I would appreciate it if the images of the patient's anterior segment and the ophthalmic examination report could be replaced with high-resolution images.

3. If you could indicate which postoperative day the anterior segment photograph of the second patient was taken, it would greatly assist in understanding the postoperative condition

4. Would it be advisable to include limbal stem cell deficiency as a differential diagnosis? If so, could you please incorporate key differential points in the discussion section?

Author Response

  • It would be greatly appreciated if you could include a photograph taken immediately post-surgery or on the first day after surgery for the first patient. This would make it easier to understand the specific surgical techniques that were employed.

Response: We are grateful for your suggestion. We added (Figure 1D& H) of the first patient’s eyes (right & left eye respectively) in the first postoperative day as per your suggestion. We added the description of figures in Figure captions as follow:

Figure. 1: Slit lamp pictures of right (D) and left (H) eyes on the first postoperative day.”

  • If possible, I would appreciate it if the images of the patient's anterior segment and the ophthalmic examination report could be replaced with high-resolution images.

Response: We are grateful for your suggestion. We replaced the images of the patient’s anterior segment with high-resolution images.

  • If you could indicate which postoperative day the anterior segment photograph of the second patient was taken, it would greatly assist in understanding the postoperative condition.

Response: We are grateful for your suggestion. We added (Figure 3D& H) of the second patient’s eyes (right & left eye respectively) in the first postoperative day of the first operation as per your suggestion. We added the description of figures in Figure captions as follow:

Figure. 3: Slit lamp pictures of right (D) and left (H) eyes on the first postoperative day after first operation.”

  • Would it be advisable to include limbal stem cell deficiency as a differential diagnosis? If so, could you please incorporate key differential points in the discussion section?

Response: We are grateful for your suggestion. We added limbal stem cell deficiency as a differential diagnosis and added this paragraph in the discussion section as per your suggestion. Line 185-190” LSCD can be distinguished from FSMK by its underlying cause, such as mechanical or chemical trauma or other known factors contributing to LSCD.8 However, because FSMK gradually damages corneal limbal stem cells and causes conjunctivization as it progresses, the possibility that this rare disease is responsible for some LSCDs of unknown cause cannot be ruled out. Whether FSMK is a contributing factor to LSCDs will have to wait until more cases accumulate.”

Also, we added a new reference as follows:

8- Deng SX, Borderie V, Chan CC et al. Global consensus on the definition, classification, diagnosis and staging of limbal stem cell deficiency. Cornea 2019;38:364-75.

Reviewer 2 Report

Comments and Suggestions for Authors

The author reports two cases of Fuchs' superficial marginal keratitis. The manuscript is well-written, so I have no comment for the author.

The author reported two interesting Fuchs' superficial marginal keratitis (FSMK) cases. The author implied that an early diagnosis and treatment could get a dramatically better prognosis by comparing the clinical course of the two patients. Moreover, the author implied that even many surgical interventions will fail if an early intensive treatment is delayed for the patient with FSMK. Furthermore. the author showed the potential value of anterior segment OCT in diagnosing FSMK, and differentiating FSMK from other peripheral ulcerative keratitis.

Author Response

The author reports two cases of Fuchs' superficial marginal keratitis. The manuscript is well-written, so I have no comment for the author.

The author reported two interesting Fuchs' superficial marginal keratitis (FSMK) cases. The author implied that an early diagnosis and treatment could get a dramatically better prognosis by comparing the clinical course of the two patients. Moreover, the author implied that even many surgical interventions will fail if an early intensive treatment is delayed for the patient with FSMK. Furthermore. the author showed the potential value of anterior segment OCT in diagnosing FSMK, and differentiating FSMK from other peripheral ulcerative keratitis.

Response: We are grateful for the review and valuable comments. We appreciated your kind and encouraging words.

Reviewer 3 Report

Comments and Suggestions for Authors

In this manuscript, the authors describe two cases of Fuchs’ superficial marginal keratitis (FSMK). In the first case, the patient was diagnosed and treated sooner than the second patient. The first patient responded better to the treatment and did not show any signs of relapse during the follow-up timeline. The second patient’s condition continued to relapse, and needed multiple periodical surgeries. It is not clear why the second patient did not respond well to the treatments, however, the authors propose that a late diagnosis and treatment might have affected patient’s response to treatments.

Given limited information on such cases and their treatment responses, it is useful to have such case reports. Please see comments below:

The title suggests that this is a review article, however, the authors describe case reports. Please change the title to “Bilateral Fuchs’ superficial marginal keratitis diagnosis and treatment: A case report” or something similar.

Use full form of MMC at the first use.

Case 1: Instead of stating ‘our hospital’ please state the name of the hospital. Please use specific times for description. Instead of saying 1.5 years have passed since the surgery (the readers do not know what was the start and end time being referred to), please state month/year when the patient was first observed, when the treatment was performed, and when the authors state 1.5 years, what is the end time point (month/year).

Similarly, when the authors state ‘at her last visit’ for the second patient, the readers have no idea when was the last visit. Please state month and year.

Line 147: Please correct to 2nd and 4th decades of life

Was there any keratopathy observed in the first case? If not, could that be one of the reasons that the first patient responded better than the second patient?

Comments on the Quality of English Language

Some edits in the text are needed

Author Response

In this manuscript, the authors describe two cases of Fuchs’ superficial marginal keratitis (FSMK). In the first case, the patient was diagnosed and treated sooner than the second patient. The first patient responded better to the treatment and did not show any signs of relapse during the follow-up timeline. The second patient’s condition continued to relapse, and needed multiple periodical surgeries. It is not clear why the second patient did not respond well to the treatments, however, the authors propose that a late diagnosis and treatment might have affected patient’s response to treatments.

Response: We are grateful for the review and valuable comments. We have addressed each of your recommendations and added point-by-point responses below.

Given limited information on such cases and their treatment responses, it is useful to have such case reports. Please see comments below:

  • The title suggests that this is a review article, however, the authors describe case reports. Please change the title to “Bilateral Fuchs’ superficial marginal keratitis diagnosis and treatment: A case report” or something similar.

Response: We are grateful for your suggestion. We changed the title to “Bilateral Fuchs’ superficial marginal keratitis diagnosis and treatment: case report.” as per your suggestion.

  • Use full form of MMC at the first use.

Response: We are grateful for your comment. We used full name of MMC at the first use in the abstract and in line 69 as per your suggestion.

  • Case 1: Instead of stating ‘our hospital’ please state the name of the hospital. Please use specific times for description. Instead of saying 1.5 years have passed since the surgery (the readers do not know what was the start and end time being referred to), please state month/year when the patient was first observed, when the treatment was performed, and when the authors state 1.5 years, what is the end time point (month/year).

Response: We are grateful for your comment. We added Nagasaki university hospital instead of stating ‘our hospital’ as per your suggestion.

Also, we added the month/year when the patient first observed, when the     treatment was performed, and when we stated 1.5 years in the manuscript as per your suggestion.

  • Similarly, when the authors state ‘at her last visit’ for the second patient, the readers have no idea when was the last visit. Please state month and year.

Response: We are grateful for your suggestion. We corrected the descriptions of ‘at her last visit’ for the second patient in the manuscript as per your suggestion.

  • Line 147: Please correct to 2nd and 4th decades of life

Response: We are grateful for your comment. We corrected it in the manuscript according to your suggestion.

  • Was there any keratopathy observed in the first case? If not, could that be one of the reasons that the first patient responded better than the second patient?

Response: We are grateful for your comment. The first case had also superficial punctate keratopathy but in a mild form than the second case. We are not sure that the severity of keratopathy has a rule in prognosis because of rarity of cases.

Reviewer 4 Report

Comments and Suggestions for Authors

This is an interesting case study of two patients with Fuchs’ superficial marginal keratitis treated. The authors showed that its progression can be suppressed by surgery and anti-inflammatory drugs. The description of the case studies is quite brief and can be expanded. The discussion section is well written stating the current understanding of the condition and its differential diagnosis, but it should provide greater context to the current two case studies. Minor comment: abbreviations such as Rt and Lt should be written in full.

Author Response

This is an interesting case study of two patients with Fuchs’ superficial marginal keratitis treated. The authors showed that its progression can be suppressed by surgery and anti-inflammatory drugs. The description of the case studies is quite brief and can be expanded. The discussion section is well written stating the current understanding of the condition and its differential diagnosis, but it should provide greater context to the current two case studies.

Response: We are grateful for the review and valuable comments. We appreciated your kind and encouraging words. Due to the rarity of FSMK cases, the available literature on the topic is very limited. In our study, we aimed to shed more light on FSMK to ensure that readers do not overlook these cases.

Minor comment: abbreviations such as Rt and Lt should be written in full.

Response: We are grateful for your comment. We corrected it in the manuscript according to your suggestion.

Round 2

Reviewer 1 Report

Comments and Suggestions for Authors

The response to the reviewer's questions seems to be well-prepared.